# Relative fat mass and cardiovascular risk in Peruvian adults: Findings from a national survey

**José A. Chaquila[1], Akram Hernández-Vásquez** **[2]\*, Jamee Guerra Valencia[3],
Fresia Miranda-Torvisco[1], Gianella Ramirez-Jeri[1]**

**1** Grupo de investigación en Nutrición Pública y Seguridad Alimentaria Nutricional, Universidad San Ignacio de Loyola, Lima, Peru, **2** Centro de Excelencia en Investigaciones Económicas y Sociales en Salud, Vicerrectorado de Investigación, Universidad San Ignacio de Loyola, Lima, Peru, **3** Facultad de Ciencias de la Salud, Universidad Privada del Norte, Lima, Peru

\* ahernandez@usil.edu.pe

## Abstract

### Background

In the Region of the Americas, particularly in low- and middle-income countries such as Peru, cardiovascular disease (CVD) remains one of the leading causes of mortality. A positive association has also been described between body fat percentage and CVD risk.

### Objective

To evaluate the association between obesity, defined by Relative Fat Mass (RFM, an anthropometric indicator that estimates total body fat), and 10-year cardiovascular risk estimated by the Framingham risk score.

### Methods

A cross-sectional study was conducted using data from the Food and Nutrition Surveillance by Life Stages survey (2017–2018) in Peru. Obesity was the exposure variable defined by RFM. The RFM was also analyzed both as a continuous and categorical variable. Generalized linear models of the gamma family with a logarithmic link were applied and stratified by sex.

### Results

Data from 651 adults were analyzed. The prevalence of obesity was 78.2% in women and 42.7% in men. After adjusting for age, poverty, fruit and vegetable consumption, and altitude of residence, obesity defined by RFM was associated with higher estimated Framingham risk scores in both sexes (Women: β: 0.48; 95% CI: 0.32–0.63; Men: β: 0.39; 95% CI: 0.23–0.56. Similar results were observed when RFM was analyzed as a continuous variable and in tertiles.

**Data availability statement:** The dataset is accessible through the Peruvian National Open Data Platform (https://www.datosabiertos.gob.pe/dataset/estado-nutricional-en-adultos-de-18-59-a%C3%B1os-per%C3%BA-2017-%E2%80%93-2018). Similarly, ENAHO data are publicly available at https://proyectos.inei.gob.pe/microdatos/.

**Funding:** The author(s) received no specific funding for this work.

**Competing interests:** The authors have declared that no competing interests exist.

## Conclusion

Obesity defined by RFM was positively associated with estimated 10-year cardiovascular risk in both sexes, with stronger association in women. These results suggest that RFM may serve as a useful tool for assessing estimated 10-year cardiovascular risk, with implications for the design of public health interventions in Peru.

## Introduction

Cardiovascular diseases (CVDs) represent one of the main challenges for health systems worldwide due to their high morbidity, mortality, disability, and economic burden [1]. In 2019, CVDs accounted for approximately 18.6 million deaths globally [2], with nearly 80% of these occurring in low- and middle-income countries [3]. In Latin America, the total number of prevalent CVDs cases more than doubled between 1990 and 2021, rising from 20 million to 47 million during this period [1]. Although Peru is among the countries with the lowest cardiovascular mortality rates in the region [4], nearly one-third of its population met no more than two out of five markers of cardiovascular health [5].

Obesity is an independent risk factor for several comorbidities, including hypertension and dyslipidemia [6]. A direct relationship has been described between body fat percentage, particularly visceral fat, and CVDs risk [7,8]. Although reference techniques such as dual-energy X-ray absorptiometry (DEXA) can assess body adiposity, their high costs and infrastructure requirements limit their use in clinical and population-based settings [9]. For this reason, anthropometric indicators remain valuable in both clinical practice and population studies because they characterize body composition and are associated with health outcomes [10], while being inexpensive and easy to obtain. Among these, the Relative Fat Mass (RFM) has emerged as an alternative anthropometric marker for obesity diagnosis. RFM has been validated against DEXA and is calculated from sex, height, and waist circumference, providing a practical estimate of body fat percentage [11]. Moreover, its predictive discrimination for heart disease mortality (0.64 in women and 0.67 in men) has been shown to be higher than that of body mass index (BMI) (0.53 and 0.56) and waist circumference (0.60 and 0.64) [12]. In addition, compared to BMI and waist circumference (WC), RFM has demonstrated a greater predictive discrimination for heart disease mortality in both sexes. Likewise, the proportion of preventable heart disease mortality attributed to RFM-diagnosed obesity was higher than that observed when using BMI or WC [12]. These findings add to previous evidence that also supports the use of RFM as a useful tool for predicting cardiovascular risk [13].

Several tools are currently available to estimate CVDs risk in adults, with risk scores being particularly useful for stratification. Among them, the Framingham risk score [14] is one of the most widely used, allowing the estimation of 10- and 30-year cardiovascular risk based on lifestyle and clinical predictors [15]. It is also recommended by the Peruvian Ministry of Health for assessing 10-year cardiovascular risk [16]. Previous studies have reported associations between the Framingham risk

score and anthropometric indicators of obesity [17], however these were conducted in populations with different ethnic backgrounds than Peru. Considering that differences in body composition and ethnicity modulate cardiovascular risk association [18], findings from high-income countries or predominantly Caucasian populations may not represent Latin American or Andean contexts. Given that CVD-related mortality in Peru increased by more than 70% between 2017 and 2022 [19], it is important to improve methods for predicting CVDs. To our knowledge, no study has evaluated the association between obesity defined by RFM and estimated 10-year cardiovascular risk using the Framingham score in a Peruvian population. Therefore, this study aimed to evaluate the association between obesity defined by RFM, considered under different operationalization approaches (binary, continuous, and tertile-based) and 10-year cardiovascular risk stratified by sex, as estimated by the Framingham risk score in Peruvian adults who participated in a national survey during 2017–2018. The present study advances knowledge beyond prior international RFM studies by comparing multiple operationalizations of RFM (binary, continuous, and tertiles) and applying sex-stratified analyses within a complex survey design in a nationally representative Latin American population, providing novel insights into optimal analytical strategies for RFM-estimated cardiovascular risk assessment in ethnically diverse contexts.

## Materials and methods

### Study type and design

An analytical cross-sectional study was conducted using the data from the 2017–2018 Food and Nutrition Surveillance by Life Stages (VIANEV) survey. The VIANEV survey has national representativeness and includes data collected on anthropometric measurements, lifestyle habits, and biochemical analyses of Peruvian adults aged 18–59 years, with a fasting period of at least 9–12 hours [20]. The survey excluded adults who consumed food prior to biochemical assessments, those with gastrointestinal diseases affecting diet, and those with anatomical conditions preventing accurate anthropometric measurement. Further methodological details of the survey are available in the official technical report [20].

The survey collected information across three domains: Metropolitan Lima (the capital of Peru), the rest of the urban area and rural areas, using a stratified, multistage, probabilistic, and independent sampling design. Sampling was conducted in two stages: first, by randomly selecting clusters as the primary sampling unit, and subsequently by randomly selecting households with adult members as the primary unit. The VIANEV 2017–2018 survey was conducted among a subsample of adults who also participated in the National Household Survey 2017 (ENAHO). ENAHO constitutes one of the largest population-based surveys in Peru and widely used to inform health decision-making and guide public policy in the country. In contrast to ENAHO, which focuses primarily on demographic and economic aspects, VIANEV participants were asked a broader set of health-related questions [21].

### Study population

Adults aged 30–59 years were included in the present analysis. Participants younger than 30 years were excluded because the development and validation of the Framingham risk score and the estimation of 10-year cardiovascular risk were performed in adults aged 30–74 years [14]. Therefore, estimating 10-year cardiovascular risk in younger individuals would not reflect a real risk [22]. Additionally, participants with missing values in any of the variables of interest were excluded.

### Variables

**Exposure variable: obesity.** Obesity was defined using the RFM index, calculated from height (cm), WC (cm), and sex of participants. WC was measured using a 200-cm measuring tape with 1-mm precision, positioning it at the midpoint between the lower margin of the last rib and the upper border of the iliac crest. Height was assessed with a fixed wooden stadiometer [20]. The following calculation was performed [11]:

$$\text{RFM}: 64 - (20 \times (\text{height}/\text{waist circumference})) + (12 \times \text{sex})$$

where sex was coded as 0 for men and 1 for women. Cut-off points of ≥40% for women and ≥30% for men were applied to define obesity as a dichotomous variable (No/Yes). The selection of these cut-off points was based on prior validation studies for obesity diagnosis and mortality prediction [23], as well as their use in a Peruvian population-based survey [24].

**Outcome variable: cardiovascular risk.** The 10-year cardiovascular risk was analyzed as a continuous variable and estimated using the Framingham risk score [14]. This score was proposed by D'Agostino et al. in 2008 using data from adults enrolled in the Framingham Heart Study. For the development of this score, eligible participants were those who attended examination cycles conducted between 1968 and 1987, had high-density lipoprotein cholesterol measurements available, and were free of CVDs at baseline. From the initial assessment, the incidence of cardiovascular events was monitored, and the risk factors that significantly predicted these events were evaluated to construct a points-based score for estimating 10-year cardiovascular risk. The score assigns risk points through an algorithm that incorporates previously identified factors: age (years), sex, systolic blood pressure, high-density lipoprotein cholesterol, total cholesterol, smoking status, diabetes diagnosis (defined as fasting glucose ≥126 mg/dL or use of diabetes-related medication), and use of antihypertensive medication. Higher scores indicate greater risk of cardiovascular events within 10 years [14]. The score was calculated using the *framingham* command in Stata SE version 18.

**Covariates.** The following variables were considered as potential confounders: age (30–39, 40–49, and 50–59 years); educational attainment (up to primary, secondary, and higher) [25]; daily sedentary time (<7 hours and ≥8 hours) [26]; household poverty status (yes/no) based on household expenditure; daily consumption of five or more servings of fruits and vegetables (yes/no) [27]; alcohol consumption in the past 30 days (yes/no); household residence (urban/rural) and altitude residence (0–499, 500–2499, and ≥2500 meters above sea level (masl) [24]. The selection of these variables was based on epidemiological criteria.

## Statistical analysis

All statistical analyses were performed using Stata version 18 (StataCorp LLC). To obtain additional socioeconomic information of VIANEV participants, the VIANEV 2017–2018 database was merged with the ENAHO 2017 survey using strata, clusters, dwellings, and households as linkage variables.

Given the characteristics of the VIANEV survey, the analyses accounted for sample weights and the complex survey design. All analyses were stratified by sex to account for sex-specific differences in body composition. Participant characteristics were described using absolute frequencies, weighted frequencies (with 95% confidence intervals [95% CI]), and measures of central tendency (means and medians) with their respective measures of dispersion (standard deviation and interquartile range).

To assess the association between RFM and the Framingham risk score, crude and adjusted generalized linear models (GLMs) from the gamma family with a logarithmic link function were employed. All covariates were included in the adjusted models based on their epidemiological relevance. This model was chosen because it is appropriate for positively skewed continuous outcomes such as Framingham score values. The strength of association was expressed as β coefficients with their 95% CI and standard errors. Additionally, as a sensitivity analysis, GLMs were also fitted considering RFM as a continuous variable and categorized into tertiles derived from our own sample, with the highest tertile representing the greatest RFM values.

## Ethical considerations

The VIANEV 2017–2018 survey data are publicly available and do not contain information that allows identification of participants. The dataset was accessed for research purposes on April 29, 2025, through the Peruvian National Open Data Platform (https://datosabiertos.gob.pe/dataset/estado-nutricional-en-adultos-de-18-59-a%C3%B1os-per%C3%BA-2017-%E2%80%93-2018). Similarly, ENAHO data are publicly available at https://proyectos.inei.gob.pe/microdatos/. At no

point did the authors have access to identifiable information about the participants. The interviewees provided their verbal consent to participate. Therefore, approval from an institutional ethics committee was not required.

## Results

A total of 651 participants were included, of whom 58.2% were women. Among men, those under 39 years of age predominated, whereas among women, the majority were aged 40–49 years. In both sexes, higher education, sedentary behavior, and non-poor households were more prevalent. Regarding the daily consumption of five or more servings of fruits and vegetables, men reported a higher intake. In addition, most households were located in urban areas and at altitudes below 500 masl (Table 1).

The prevalence of obesity was higher in women than in men, whereas the median estimated 10-year cardiovascular risk in men was more than three times higher than that observed in women (Table 2). In both sexes, a higher prevalence of obesity and higher estimated Framingham risk scores were observed with increasing age. Among men, obesity prevalence and Framingham risk scores were higher in those with greater educational attainment, whereas among women obesity prevalence was lower in those with higher education. In men, obesity prevalence and mean Framingham risk scores decline with increasing altitude, while no consistent pattern is observed among women. Additionally, when comparing the Framingham risk score between non-obese and obese participants, non-obese men (7.65) had a lower mean Framingham risk score than obese men (13.75) (mean difference: −6.11, 95% CI: −8.63 to −3.58, p < 0.001). Similarly, non-obese women (2.07) had a lower Framingham risk score than obese women (4.01) (mean difference: −1.95, 95% CI: −2.63 to −1.27, p < 0.001).

The association between obesity defined by RFM and the estimated 10-year Framingham risk score was positive in both the crude and adjusted models for both sexes. Regarding obesity, the adjusted models showed a higher coefficient in women. Additionally, to assess whether RFM retained the direction and significance of its association when analyzed as a continuous variable with the Framingham risk score, a sensitivity analysis was conducted, showing a similar positive coefficient in both sexes (Table 3).

## Discussion

The objective of the present study was to evaluate the association between obesity defined by RFM and the estimated 10-year cardiovascular risk according to the Framingham risk score in Peruvian adults aged 30–59 years. A positive and significant association was observed in both sexes. Furthermore, it was found that for each unit increase in RFM, the estimated 10-year cardiovascular risk increased by 0.07 in men and 0.06 in women on the logarithmic scale, corresponding to approximately 7% and 6% higher estimated risk, respectively. This indicates that higher estimated total body fat percentage was associated with proportionally higher estimated 10-year cardiovascular risk. To the best of our knowledge, this is the first study to evaluate the association between RFM, an alternative estimator of total body fat percentage, and the estimated 10-year cardiovascular risk using the Framingham score.

### Comparison with previous studies

The findings of this study revealed that the presence of obesity defined by RFM was associated with a 48% and 61% higher estimated 10-year cardiovascular risk in men and women, respectively, compared with non-obese individuals. Additionally, when stratifying by RFM tertiles, a dose-response relationship was observed, with the estimated 10-year cardiovascular risk in the highest tertile being 84% in men and 60% in women, compared with the lowest tertile. These results are consistent with previous evidence supporting the clinical utility of RFM in assessing cardiovascular risk, both in cross-sectional and prospective studies. For instance, in a cross-sectional analysis of more than 11,000 Chinese adults, each standard deviation increase in RFM was associated with higher odds of CVDs in men (OR: 1.66; 95% CI: 1.36–2.02) and women (OR: 1.26; 95% CI: 1.08–1.47) [28]. Similarly, in U.S. adults, RFM was significantly associated with CVDs in

**Table 1. Sample characteristics by sex in adults aged 30–59 years, VIANEV Survey.**

| Characteristics | Male | | Female | |
|---|---|---|---|---|
| | n | % weighted | n | % weighted |
| **Age group (years)** | | | | |
| 30–39 | 96 | 36.9 (30.6-43.6) | 135 | 34.6 (29.1-40.5) |
| 40–49 | 84 | 29.8 (24.2-36.0) | 136 | 37.2 (31.6-43.3) |
| 50–59 | 92 | 33.4 (27.2-40.2) | 108 | 28.2 (23.6-33.3) |
| **Educational level** | | | | |
| Up to Primary | 60 | 17.7 (13.2-23.4) | 126 | 25.3 (20.8-30.3) |
| Secondary | 116 | 39.9 (32.9-47.4) | 130 | 34.9 (29.3-40.8) |
| Higher | 96 | 42.4 (35.3-49.7) | 123 | 39.9 (34.0-46.1) |
| **Sitting time** | | | | |
| Up to 7 hours | 220 | 78.6 (71.7-84.2) | 338 | 87.6 (82.8-91.2) |
| 8 hours or more | 52 | 21.4 (15.8-28.3) | 41 | 12.4 (8.8-17.2) |
| **Household in poverty** | | | | |
| No | 223 | 83.7 (77.8-88.2) | 315 | 84.9 (80.5-88.5) |
| Yes | 49 | 16.3 (11.8-22.2) | 64 | 15.1 (11.5-19.5) |
| **Fruit and vegetable intake (≥5 servings/day)** | | | | |
| No | 191 | 68.4 (61.1-74.8) | 287 | 73.4 (67.4-78.6) |
| Yes | 81 | 31.6 (25.2-38.9) | 92 | 26.6 (21.4-32.6) |
| **Alcohol consumption** | | | | |
| No | 109 | 32.8 (27.1-39.1) | 230 | 56.4 (50.2-62.5) |
| Yes | 163 | 67.2 (60.9-72.9) | 149 | 43.6 (37.5-49.8) |
| **Residence area** | | | | |
| Rural | 110 | 24.6 (20.8-28.8) | 119 | 17.5 (14.7-20.6) |
| Urban | 162 | 75.4 (71.2-79.2) | 260 | 82.5 (79.4-85.3) |
| **Altitude residence (masl)** | | | | |
| 0–499 | 176 | 73.6 (67.7-78.7) | 272 | 74.2 (68.4-79.3) |
| 500–2499 | 44 | 11.1 (7.9-15.2) | 51 | 11.4 (7.9-16.0) |
| 2500 or higher | 52 | 15.3 (10.9-21.1) | 56 | 14.4 (10.3-19.8) |

masl.: meters above sea level.

All estimates accounted for the VIANEV sample design.

both sexes, with 4% higher odds in men and 3% higher odds in women when comparing the highest versus the lowest quintile of RFM [29]. Moreover, prospective studies have documented that higher RFM is associated with greater incidence of heart failure [30], as well as cardiovascular mortality [12]. Despite variability in the magnitude of the association between RFM and CVDs across studies, which may be attributed to differences in how outcomes are measured and defined, taken together, these findings support the potential clinical utility of RFM in cardiovascular risk assessment.

## Potential explanatory mechanisms

The observed association between RFM and estimated CVDs risk may reflect both pathophysiological and methodological factors. RFM incorporates height and WC in estimating body fat percentage [11]. Waist circumference is a marker of central adiposity distribution, particularly visceral adiposity [31], which has been widely associated with cardiovascular risk factors [32]. Visceral adipose tissue is metabolically active and characterized by a more lipolytic and inflammatory profile than subcutaneous adiposity, and is associated with lipid and glucose metabolism disturbances [33]. Greater visceral

**Table 2. Relative Fat Mass and the estimated 10-year Framingham Risk Score by sex and sociodemographic and lifestyle characteristics in Peruvian adults aged 30–59 years, VIANEV Survey.**

| Characteristics | Male | | | | Female | | | |
|---|---|---|---|---|---|---|---|---|
| | Relative Fat Mass | | Framingham risk score | | Relative Fat Mass | | Framingham risk score | |
| | Mean | High | Mean | Median | Mean | High | Mean | Median |
| | SD | % | SD | (p25, p75) | SD | % | SD | (p25, p75) |
| Overall | 29.3 (4.2) | 42.7 | 10.3 (9.1) | 7.5 (3.9-13.4) | 43.2 (3.9) | 78.2 | 3.6 (3.9) | 2.2 (1.3-4.1) |
| **Age group (years)** | | | | | | | | |
| 30–39 | 28.2 (3.8) | 31.8 | 4.8 (3.9) | 3.6 (2.4-5.6) | 42.3 (4.7) | 69.7 | 1.3 (0.9) | 1.1 (0.7-1.6) |
| 40–49 | 29.7 (4.8) | 43.1 | 8.6 (5.9) | 7.3 (4.5-10.3) | 43.5 (3.8) | 81.6 | 3.1 (2.9) | 2.4 (1.8-3.3) |
| 50–59 | 30.3 (3.7) | 54.4 | 17.7 (10.1) | 14.9 (10.2-22.4) | 43.9 (3.5) | 84.1 | 7.0 (5.1) | 5.4 (3.5-9.2) |
| **Educational level** | | | | | | | | |
| Up to Primary | 26.9 (4.4) | 16.6 | 9.3 (7.1) | 8.5 (3.9-11.5) | 44.4 (3.9) | 89.8 | 3.9 (4.7) | 3.0 (1.7-4.3) |
| Secondary | 29.2 (4.3) | 40.6 | 9.9 (9.7) | 6.6 (3.1-13.1) | 43.6 (4.1) | 82.1 | 3.3 (3.6) | 2.0 (1.2-3.9) |
| Higher | 30.5 (3.4) | 55.6 | 11.0 (8.6) | 8.0 (4.9-13.4) | 42.1 (3.5) | 67.3 | 3.6 (3.8) | 2.1 (1.3-4.2) |
| **Sitting time** | | | | | | | | |
| Up to 7 | 29.1 (4.1) | 41.4 | 10.4 (9.6) | 7.3 (3.8-13.0) | 43.3 (4.0) | 79.7 | 3.5 (3.9) | 2.4 (1.3-4.1) |
| 8 or more | 30.2 (4.4) | 47.6 | 9.8 (6.2) | 8.0 (4.6-14.3) | 42.1 (3.9) | 67.4 | 3.9 (4.5) | 1.8 (1.3-3.8) |
| **Household in poverty** | | | | | | | | |
| No | 29.8 (3.9) | 45.7 | 10.8 (9.1) | 7.9 (4.0-14.7) | 43.1 (4.0) | 76.6 | 3.7 (4.1) | 2.2 (1.3-4.1) |
| Yes | 27.1 (4.7) | 27.4 | 7.2 (7.1) | 6.1 (2.8-8.6) | 43.7 (3.9) | 87.2 | 3.0 (2.8) | 2.2 (1.1-3.9) |
| **Fruit and vegetable intake** | | | | | | | | |
| No | 29.5 (4.2) | 45.9 | 10.1 (8.6) | 7.4 (3.9-12.9) | 43.1 (4.1) | 77.6 | 3.7 (4.3) | 2.4 (1.3-4.1) |
| Yes | 29.0 (3.9) | 35.8 | 10.7 (9.5) | 7.6 (3.6-14.0) | 43.3 (3.5) | 79.7 | 3.2 (3.0) | 2.1 (1.2-3.9) |
| **Alcohol consumption** | | | | | | | | |
| No | 29.0 (4.7) | 41.9 | 10.7 (8.9) | 8.5 (4.5-13.1) | 43.5 (3.7) | 83.0 | 3.4 (3.6) | 2.1 (1.4-4.1) |
| Yes | 29.5 (3.9) | 43.1 | 10.1 (8.8) | 6.8 (3.6-13.7) | 42.7 (4.1) | 71.9 | 3.8 (4.3) | 2.4 (1.2-3.9) |
| **Residence area** | | | | | | | | |
| Rural | 27.3 (4.8) | 25.8 | 7.8 (8.1) | 6.0 (3.4-10.0) | 43.2 (4.9) | 83 | 3.4 (5.2) | 2.5 (1.3-3.8) |
| Urban | 30.0 (3.6) | 48.2 | 11.1 (8.5) | 8.0 (4.0-14.7) | 43.2 (3.7) | 77.2 | 3.6 (3.6) | 2.2 (1.3-4.3) |
| **Altitude residence (masl)** | | | | | | | | |
| 0–499 | 30.0 (3.7) | 46.8 | 10.8 (8.6) | 7.9 (4.0-14.7) | 43.1 (4.1) | 76.6 | 3.8 (4.2) | 2.2 (1.3-4.5) |
| 500–2499 | 27.0 (5.4) | 21.1 | 10.0 (12.9) | 6.5 (3.3-12.6) | 43.8 (3.2) | 88.4 | 2.7 (2.2) | 2.2 (1.1-3.9) |
| 2500 or higher | 27.9 (4.3) | 38.4 | 7.7 (5.8) | 6.8 (4.6-8.9) | 43.0 (3.8) | 78.5 | 3.4 (3.2) | 2.4 (1.4-3.9) |

All estimates accounted for the VIANEV sample design. p25: 25th percentile, p75: 75th percentile, SD: standard deviation, masl: meters above sea level.

adiposity has also been associated with higher blood pressure [34]. Consistent with this mechanistic framework, a longitudinal study in Peruvian adults reported that RFM-defined obesity was associated with a higher incidence of hypertension over five years, which represents a major component of cardiovascular risk, thereby supporting the relevance of RFM in cardiometabolic risk assessment [35]. Taken together, increased central adiposity has been closely related to several components of cardiovascular risk [14]. Consistent with this, it has been documented that for every 10 cm increase in WC, CVDs risk increases by 3.4% in women and 4% in men [36]. From a methodological perspective, RFM has been validated against DEXA, the gold standard for assessing body composition [9]. Furthermore, RFM is based on the height-to-waist ratio, essentially the inverse of the WHtR, which has shown a strong association with CVDs risk [36]. However, our results should be interpreted with caution as the Framingham risk score has neither been validated nor recalibrated for the

**Table 3. Associations between Relative Fat Mass and the estimated 10-year Framingham Risk Score by sex, VIANEV Survey.**

| | Generalized linear model (GLM) of the gamma family | | | | |
| --- | --- | --- | --- | --- | --- |
| | Crude | | | Adjusted* | |
| Model | β (95% CI) | SE | | β (95% CI) | SE |
| **Male** | | | | | |
| Obesity by Relative Fat Mass | | | | | |
| No | Reference | | | Reference | |
| Yes | 0.59 (0.37-0.81) | 0.11 | | 0.39 (0.23-0.56) | 0.08 |
| **Female** | | | | | |
| Obesity by Relative Fat Mass | | | | | |
| No | Reference | | | Reference | |
| Yes | 0.66 (0.42-0.91) | 0.12 | | 0.48 (0.32-0.63) | 0.08 |
| **Male** | | | | | |
| Tertiles by Relative Fat Mass | | | | | |
| Low | Reference | | | Reference | |
| Medium | 0.57 (0.30-0.84) | 0.14 | | 0.39 (0.19-0.59) | 0.10 |
| High | 0.83 (0.56-1.09) | 0.13 | | 0.61 (0.42-0.80) | 0.10 |
| **Female** | | | | | |
| Tertiles by Relative Fat Mass | | | | | |
| Low | Reference | | | Reference | |
| Medium | 0.32 (0.05-0.60) | 0.14 | | 0.19 (0.03-0.35) | 0.08 |
| High | 0.68 (0.41-0.95) | 0.14 | | 0.47 (0.28-0.67) | 0.10 |
| **Male** | | | | | |
| Relative Fat Mass | 0.10 (0.07-0.12) | 0.01 | | 0.07 (0.05-0.09) | 0.01 |
| **Female** | | | | | |
| Relative Fat Mass | 0.06 (0.03-0.08) | 0.01 | | 0.06 (0.04-0.08) | 0.01 |

SE: standard error, CI: confidence intervals, β: coefficient. All estimates accounted for the VIANEV sample design. * Adjusted for age group, area of residence, educational level, sitting time, poverty status, fruit and vegetable intake, alcohol consumption and altitude of residence.

Peruvian population. Chile is one of the few countries in the region that has undertaken efforts to recalibrate the model for its own population [37]. Since the score was originally developed using baseline cardiovascular event rates from a U.S. cohort, the absolute levels of predicted 10-year risk may differ substantially from those in the Peruvian context and other Andean countries. Nevertheless, the model likely preserves the relative ranking of individuals according to higher or lower estimated cardiovascular risk.

## Sex-based differences

Additionally, our findings show a stronger association between obesity and estimated cardiovascular risk in women, which is consistent with evidence indicating a higher burden of cardiovascular risk factors in women compared to men [38]. In Peru, studies have consistently reported a higher prevalence of obesity and metabolic syndrome among women, who also have markedly higher odds of abdominal obesity, reinforcing sex-specific vulnerability in adiposity-related cardiometabolic risk [39,40]. Together, these findings may help contextualize why increases in RFM are associated with larger relative changes in estimated 10-year cardiovascular risk among women in our study.

On the other hand, epidemiological studies in Latin America [4] have reported a higher incidence of CVDs in men than in women (655 vs. 531 per 100,000 individuals), and additional evidence indicates that Peruvian men present worse cardiovascular health than women [5,41]. Of note, compared with previous reports, our results address related but not

identical dimensions of cardiovascular risk. Evidence from other Andean populations shows that despite women having a higher prevalence of risk factors, men experienced higher cardiovascular event incidence, suggesting that estimated risk may not fully reflect observed outcomes [42]. This pattern suggests that a greater burden of risk factors does not necessarily translate into higher event incidence, and that sex-specific patterns of estimated cardiovascular risk may not align perfectly with observed outcomes. These findings do not contradict our results, as the Framingham Risk Score estimates predicted cardiovascular risk based on multiple factors, which may vary independently of actual event incidence. While our study assessed estimated 10-year risk, previous studies examined observed cardiovascular events. Both approaches are complementary and highlight the need for longitudinal studies in Peru to determine whether adiposity has differential cardiometabolic effects by sex and how these translate into actual outcomes.

## Implications for public health and clinical practice

In the Region of the Americas, particularly in low- and middle-income countries such as Peru, CVDs remain one of the leading causes of mortality. In response, and within the framework of the Global Action Plan for the Prevention and Control of Noncommunicable Diseases [43], the World Health Organization has promoted the Global HEARTS Initiative [44], and its regional adaptation, HEARTS in the Americas, which aims to strengthen health service performance for CVDs prevention and control. A main component of this strategy is the estimation of 10-year cardiovascular risk and the use of simple, accessible tools for its calculation in resource-limited settings [45]. Given that the use of simple anthropometric indicators is key in these strategies, it is relevant to evaluate their relationship with cardiovascular risk estimates in local populations. In this context, our findings suggest that obesity defined by RFM is associated with higher 10-year cardiovascular risk, as estimated by the Framingham risk score, which may be useful for future research exploring the potential role of RFM as an anthropometric marker in cardiovascular risk stratification. Although BMI is widely used to define obesity and is also associated with CVDs [46], it has important limitations that may lead to misclassification when detecting adults with excess body fat [47]. A high BMI does not necessarily reflect greater adiposity, as it is derived from total body weight and does not differentiate by sex [48]. These findings seek to contribute to the growing body of evidence supporting the need to implement alternative diagnostic criteria for obesity that can more accurately estimate cardiometabolic risk in Peru. To advance this effort, several key steps are required. First, drawing on the experience of countries such as the United States [49] and the United Kingdom [50]—where WC and WHtR measurements are endorsed by clinical guidelines for estimated cardiovascular risk assessment—there is a need to recalibrate RFM for the Peruvian context, given the country's distinct ethnic characteristics. This requires population-based studies that include a valid reference standard for adiposity, such as DEXA, as well as cardiometabolic outcomes that can inform the establishment of appropriate cut-off points. Subsequently, considering the simplicity and very low cost of measuring WC and height in primary care settings, implementing RFM through pilot studies would help quantify the cardiovascular risk that BMI commonly underestimates. The main barriers to implementation would likely include the need to update clinical guidelines and potential resistance among clinical personnel when adopting a new adiposity estimator.

## Study limitations

This study has several limitations that should be considered when interpreting our findings. The analysis was based on secondary data from the VIANEV 2017–2018 survey; therefore, recording errors or missing information cannot be ruled out, as the data were collected for purposes other than this research. In addition, due to the cross-sectional design of the study, it is not possible to establish causality between exposure and outcome. Nevertheless, these results may represent an initial step toward the development of more complex methodologies aimed at identifying potential causal pathways between variables. Moreover, the characteristics of excluded participants may have influenced the strength of the reported associations. We also conducted a sensitivity analysis to evaluate whether the association retained its direction and significance when modifying the nature of the exposure variable (RFM). Furthermore, the measurement of anthropometric

variables, although standardized, may vary across interviewers or equipment. Some self-reported variables, such as lifestyle habits or living conditions, may have been influenced by social desirability or recall bias. The lack of potentially confounding variables, such as diet quality, energy intake or more specific physical activity measurement, could also result in residual confounding, affecting the magnitude and direction of the estimated associations due to its strong influence on body fat percentage and components of the Framingham Risk Score; nevertheless, the adjustment models included indirect measures of diet quality (daily consumption of five or more portions of fruits and vegetables) and physical activity (daily hours of sedentary behavior) in order to reduce confounding, although residual confounding cannot be completely excluded. Although the Framingham risk score is widely used to estimate 10-year cardiovascular risk, its accuracy when applied outside the original U.S. population may lead to over- or underestimation of risk. In addition, the lack of validation in the Peruvian population may limit its applicability in this context. Nevertheless, the score is currently recommended by the Peruvian Ministry of Health for cardiovascular risk stratification [16], which supports its use in local epidemiological analyses. Finally, since the data were collected before the COVID-19 pandemic, patterns of obesity and cardiovascular risk may have shifted thereafter, restricting the direct applicability of our findings to the current context.

## Conclusion

RFM was positively associated with the estimated 10-year Framingham risk score in Peruvian adults of both sexes, with a stronger association observed in women than in men. We recommend that future studies evaluate the interchangeability of RFM with commonly used anthropometric markers for estimating cardiovascular risk, in order to identify the simplest and most effective indicator for estimated cardiovascular risk assessment in the Peruvian population.

## Acknowledgments

We express our gratitude to the National Institute of Statistics and Informatics of Peru and the National Center for Food and Nutrition of Peru for conducting the data collection and for making these data publicly available, thereby enabling research that contributes to the improvement of public health policies in the country.

## Author contributions

**Conceptualization:** José A. Chaquila, Jamee Guerra Valencia, Fresia Miranda-Torvisco, Gianella Ramirez-Jeri.

**Data curation:** Akram Hernández-Vásquez.

**Formal analysis:** Akram Hernández-Vásquez.

**Methodology:** José A. Chaquila, Akram Hernández-Vásquez, Jamee Guerra Valencia.

**Validation:** José A. Chaquila, Jamee Guerra Valencia.

**Visualization:** Akram Hernández-Vásquez.

**Writing – original draft:** José A. Chaquila, Akram Hernández-Vásquez, Fresia Miranda-Torvisco, Gianella Ramirez-Jeri.

**Writing – review & editing:** José A. Chaquila, Akram Hernández-Vásquez, Jamee Guerra Valencia, Fresia Miranda-Torvisco, Gianella Ramirez-Jeri.

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
