## [Decision Letter · Decision Letter 0]

12 Nov 2025

Dear Dr. Hernández-Vásquez,

Thank you for submitting your manuscript to PLOS ONE. After careful consideration, we feel that it has merit but does not fully meet PLOS ONE’s publication criteria as it currently stands. Therefore, we invite you to submit a revised version of the manuscript that addresses the points raised during the review process.

We look forward to receiving your revised manuscript.

Kind regards,

Neftali Eduardo Antonio-Villa, MD PhD

Academic Editor

PLOS ONE

Journal Requirements:

2. Please note that your Data Availability Statement is currently missing a direct link to access each database. If your manuscript is accepted for publication, you will be asked to provide these details on a very short timeline. We therefore suggest that you provide this information now, though we will not hold up the peer review process if you are unable.

3. When completing the data availability statement of the submission form, you indicated that you will make your data available on acceptance. We strongly recommend all authors decide on a data sharing plan before acceptance, as the process can be lengthy and hold up publication timelines. Please note that, though access restrictions are acceptable now, your entire data will need to be made freely accessible if your manuscript is accepted for publication. This policy applies to all data except where public deposition would breach compliance with the protocol approved by your research ethics board. If you are unable to adhere to our open data policy, please kindly revise your statement to explain your reasoning and we will seek the editor's input on an exemption. Please be assured that, once you have provided your new statement, the assessment of your exemption will not hold up the peer review process

Additional Editor Comments:

After the revision of two external referees, they identified methodological and statistical issues that the authors should addressed. Please, be aware that failing in addressing these issues appropriately may merit in withdrawing the manuscript from the review process.

Reviewers' comments:

Reviewer's Responses to Questions

**Comments to the Author**

1. Is the manuscript technically sound, and do the data support the conclusions?

Reviewer #1: Yes

Reviewer #2: Partly

2. Has the statistical analysis been performed appropriately and rigorously?

Reviewer #1: I Don't Know

Reviewer #2: No

3. Have the authors made all data underlying the findings in their manuscript fully available?

Reviewer #1: Yes

Reviewer #2: No

4. Is the manuscript presented in an intelligible fashion and written in standard English?

Reviewer #1: Yes

Reviewer #2: Yes

Reviewer #1: The manuscript addresses an important public health question: the association between relative fat mass (RFM) and 10-year cardiovascular risk in a Peruvian population. The topic is timely, especially given the high burden of obesity and CVD in low- and middle-income countries. The paper is well-organized, methodologically sound, and provides valuable findings. However, several aspects require clarification and strengthening before the manuscript can be considered for publication.

1.Novelty and Contribution

- The authors state this is the first study in Peru to link RFM with the Framingham risk score. While this is true, the novelty should be emphasized more clearly in the Introduction and Discussion. At present, the contribution is somewhat underplayed against the backdrop of existing international studies.

2. Choice of Cardiovascular Risk Tool

- The Framingham risk score has not been validated in the Peruvian population, as the authors themselves note. This limitation is significant and may affect the accuracy of the estimates. The Discussion should more critically evaluate the implications of using this score and consider whether recalibrated regional risk equations (if available) might yield different results.

3. Cross-Sectional Design

- The limitation that causality cannot be inferred should be highlighted earlier (e.g., in the Abstract or Introduction), not only in the Discussion. The authors should also consider whether reverse causality (e.g., CVD risk factors influencing body fat distribution) could bias results.

4. Definition of Obesity via RFM

- The rationale for the chosen cut-off points (≥40% for women and ≥30% for men) is based on international validation studies. However, whether these cutoffs are appropriate for the Peruvian population is uncertain. Could the authors provide sensitivity analyses using alternative thresholds or justify more strongly why these cutoffs are optimal locally?

5. Statistical Methods

-The use of generalized linear models of the gamma family with a log link is appropriate for skewed outcomes. However, the paper would benefit from an explanation of why this model was selected over alternatives (e.g., quantile regression). Additionally, please clarify whether survey weights and clustering were fully accounted for in all regression analyses.

6. Potential Confounders

-Important confounders such as alcohol consumption, physical activity beyond sedentary time, and dietary quality were not included. The absence of these variables should be acknowledged as a limitation in more detail.

7. Sex-Specific Findings

- The stronger association in women is intriguing. The Discussion offers some explanations (biological and social), but these remain speculative. The authors should expand on possible mechanisms and highlight the need for future longitudinal research to clarify causality.

8. Abstract

- The Abstract is concise, but the phrase “positively associated” could be replaced with a more precise quantitative description (e.g., “associated with a 61% higher predicted risk in women”).

9. Line 63–69: The claim that RFM’s discriminatory capacity surpasses BMI and waist circumference should be better referenced, and the cited evidence summarized more explicitly.

10. Methods

- Please specify how missing data were handled (listwise deletion vs. imputation).

- Clarify whether waist circumference was measured at the midpoint between the lowest rib and iliac crest or at another anatomical site.

11. Results

- Tables are informative but could be more reader-friendly if prevalence ratios (PRs) were included directly in Table 3 rather than only in the footnotes.

- Consider reporting absolute mean differences in Framingham scores between obese and non-obese participants for interpretability.

12. Discussion

- The section on public health implications (lines 263–280) could be expanded to comment on how RFM might realistically be integrated into Peruvian clinical practice and whether it could replace or complement BMI in national guidelines.

- Please update references to regional guidelines or ongoing WHO/PAHO initiatives to strengthen applicability.

Reviewer #2: well structured manuscipt. although, stduy have not add a new to knowledge as the topic is previousely known, the following suggestions may improve it more :

Methodology: because you retrieve a seconday data so its preferde to refered as Retreospective SURVEY correation design)

* explain the sampling processes in you maunscript (*how you select the participants within your study) as what discussed here is about the population and inclusion criteria of primary survey!

* most important is regaerding the Framingham risk score : WHEN AND HOW THE STUDY WAS APPLIED THE SCORE ? ; the scoring system must be also eplained

confusion was arised between the VIANEV and ENAHP ?? explain!

statistical analysis: median is not required as the RFM is a continous variable (Mean is enough)

correlation coefficient is required to investigate the correlation

Results : MASL abberivation (NO space )

TABLE 2; you have to include the correlation coefficient (r) and sig so the table must provide a valuable data

table 3 ; full detailed regression ,odel must be provided so the study can predicte or exclude the cofounding factors

duiscussion: well, its preffered to also add discussion about the table 2 (after you made the mentioned comments regarding the tabel 2)

conclusion: must include the main finding of table 2

**Do you want your identity to be public for this peer review?** For information about this choice, including consent withdrawal, please see our Privacy Policy

Reviewer #1: No

Reviewer #2: No

---

## [Author Response · Author response to Decision Letter 1]

9 Dec 2025

RESPONSE LETTER

Authors’ comments: We thank both reviewers for their comments. With their input and the changes made, we believe that the manuscript has improved substantially.

Reviewer #1:

The manuscript addresses an important public health question: the association between relative fat mass (RFM) and 10-year cardiovascular risk in a Peruvian population. The topic is timely, especially given the high burden of obesity and CVD in low- and middle-income countries. The paper is well-organized, methodologically sound, and provides valuable findings. However, several aspects require clarification and strengthening before the manuscript can be considered for publication.

1.Novelty and Contribution

- The authors state this is the first study in Peru to link RFM with the Framingham risk score. While this is true, the novelty should be emphasized more clearly in the Introduction and Discussion. At present, the contribution is somewhat underplayed against the backdrop of existing international studies.

Response: Thank you for the suggestion. We have added a text in the introduction acknowledging that differences in body composition and ethnicity modulate the association with cardiovascular risk

Changes: A text was added to the introduction section. Now it reads:

“Considering that differences in body composition and ethnicity modulate cardiovascular risk association [18], findings from high-income countries or predominantly Caucasian populations may not represent Latin American or Andean contexts.”

2. Choice of Cardiovascular Risk Tool

- The Framingham risk score has not been validated in the Peruvian population, as the authors themselves note. This limitation is significant and may affect the accuracy of the estimates. The Discussion should more critically evaluate the implications of using this score and consider whether recalibrated regional risk equations (if available) might yield different results.

Response: Thank you for the observation. We agree that lack of validation of the Framingham risk score in the Peruvian population may affect the accuracy of the estimates. This limitation was acknowledged in the Discussion section, where we state: “Although the Framingham score is widely used to estimate 10-year cardiovascular risk, its accuracy when applied outside the original U.S. population may lead to over- or underestimation of risk. In addition, the lack of validation in the Peruvian population may limit its applicability in this context.”

It is important to note that the Framingham risk score is endorsed by the Peruvian Ministry of Health as a reference tool for cardiovascular risk stratification, which supports its use in local clinical and epidemiological settings. We have incorporated an additional sentence acknowledging this.

Additionally, we have incorporated a text in the discussion section regarding the implications of this lack of validity in interpreting our findings

Changes: In the discussion section we added the following text:

“However, our results should be interpreted with caution as the Framingham risk score has neither been validated nor recalibrated for the Peruvian population. Chile is one of the few countries in the region that has undertaken efforts to recalibrate the model for its own population [37]. Since the score was originally developed using baseline cardiovascular event rates from a U.S. cohort, the absolute levels of predicted 10-year risk may differ substantially from those in the Peruvian context and other Andean countries. Nevertheless, the model likely preserves the relative ranking of individuals indicating who presents higher or lower estimated cardiovascular risk.”

We have added the following text to the limitations section:

“Nevertheless, the score is currently recommended by the Peruvian Ministry of Health for cardiovascular risk stratification [16], which supports its use in local epidemiological analyses”

3. Cross-Sectional Design

- The limitation that causality cannot be inferred should be highlighted earlier (e.g., in the Abstract or Introduction), not only in the Discussion. The authors should also consider whether reverse causality (e.g., CVD risk factors influencing body fat distribution) could bias results.

Response: Thank you for this observation. According to the STROBE report guidelines (https://pmc.ncbi.nlm.nih.gov/articles/PMC2034723/), the study limitations should be reported in the Discussion section. They recommend summarizing what was done and found in the Abstract, while the Introduction should present the state of the art regarding the problem and outline the objectives. In the Discussion section, the limitations of the study should indeed be addressed, as specified in the guideline. Therefore, in order to follow established reporting standards, we have kept the limitations in the Discussion section.

Regarding the possibility of reverse causality, we consider it unlikely in this context. The Framingham Risk Score represents an estimate of future cardiovascular risk rather than past or current clinical event. Thus, it cannot temporally precede or influence current body composition. For this reason, due to the nature of the Framingham Risk Score, reverse causality with the RFM—which does estimate current body fat percentage—is very unlikely. Additionally, we consider that having already stated in the abstract that our study will assess an association is sufficient for the reader’s understanding.

Changes: None

4. Definition of Obesity via RFM

- The rationale for the chosen cut-off points (≥40% for women and ≥30% for men) is based on international validation studies. However, whether these cutoffs are appropriate for the Peruvian population is uncertain. Could the authors provide sensitivity analyses using alternative thresholds or justify more strongly why these cutoffs are optimal locally?

Response: Thank you for the observation. We agree on the importance of the cut-off selection for defining obesity using the RFM. Nevertheless, as ethnic and anthropometric diversity across countries can influence body composition, cut-off points developed in other populations may not be appropriate for the Peruvian context. In line with this, the Lancet Commission on clinical obesity recommends using country-specific cut-off points when determining an obesity diagnosis (1). However, given the absence of validated RFM cut-off points for the Peruvian population and the variability of those proposed in different countries (Table A), we decided to use the original study’s cut-offs (≥40% for women and ≥30% for men). It is important to note that to address concerns regarding threshold dependence, we also conducted a sensitivity analysis treating RFM as a continuous variable, without applying a specific cut-off.

1. Rubino F, Cummings DE, Eckel RH, et al. Definition and diagnostic criteria of clinical obesity. Lancet Diabetes Endocrinol. 2025;13(3):221-262. doi:10.1016/S2213-8587(24)00316-4

Table A. Cutoff points for defining obesity according to RFM used in different countries.

Study Population study Cut-off-point

Usefulness of relative fat mass in estimating body adiposity in Korean adult population

https://www.jstage.jst.go.jp/article/endocrj/66/8/66_EJ19-0064/_article/-char/ja/

Corea ≥25 men

≥35 women

Relative fat mass and prediction of incident atrial fibrillation, heart failure and coronary artery disease in the general population

https://www.nature.com/articles/s41366-023-01380-8

Netherlands ≥26 men

≥38 women

Predictive values of relative fat mass and body mass index on cardiovascular health in community-dwelling older adults: Results from the Longevity Check-up (Lookup) 7+

https://www.maturitas.org/article/S0378-5122(24)00106-3/fulltext

Italy ≥27 men

≥40 women

Relative fat mass as an estimator of body fat percentage in Chilean adults

https://www.nature.com/articles/s41430-024-01464-2

Chile ≥22,7 men

≥32,4 women

In addition, and in response to the reviewer’s concern, we conducted an additional sensitivity analysis using a statistical criterion based on tertiles and is reported in the New Table 3. The results showed the same pattern as those obtained with the original cut-offs.

Changes: We conducted an additional sensitivity analysis using a tertile-based classification derived from our own sample. The results of this analysis have been incorporated into the revised manuscript and added to Table 3.

Table 3. Associations between Relative Fat Mass and the 10-year Framingham Risk Score by sex

Generalized linear model (GLM) of the gamma family

Crude Adjusted*

Model β (95% CIs) SE β (95% CIs) SE

Male

Obesity by Relative Fat Mass

No Reference Reference

Yes 0.59 (0.37-0.81)a 0.11 0.39 (0.23-0.56)b 0.08

Female

Obesity by Relative Fat Mass

No Reference Reference

Yes 0.66 (0.42-0.91)c 0.12 0.478 (0.32-0.63)d 0.08

Male

Tertiles by Relative Fat Mass

Low Reference Reference

Medium 0.57 (0.30-0.84) 0.14 0.39 (0.19-0.59) 0.10

High 0.83 (0.56-1.09) 0.13 0.61 (0.42-0.80) 0.10

Female

Tertiles by Relative Fat Mass

Low Reference Reference

Medium 0.32 (0.05-0.60) 0.14 0.19 (0.03-0.35) 0.08

High 0.68 (0.41-0.95) 0.14 0.47 (0.28-0.67) 0.10

Male

Relative Fat Mass 0.10 (0.07-0.12) 0.01 0.07 (0.05-0.09) 0.01

Female

Relative Fat Mass 0.06 (0.03-0.08) 0.01 0.06 (0.04-0.08) 0.01

SE: standard error, CIs: confidence intervals, β: coefficient. All estimates accounted for the VIANEV sample design. * Adjusted for age group, area of residence, educational level, sitting time, poverty status, fruit and vegetable intake, alcohol consumption and altitude of residence.

5. Statistical Methods

-The use of generalized linear models of the gamma family with a log link is appropriate for skewed outcomes. However, the paper would benefit from an explanation of why this model was selected over alternatives (e.g., quantile regression). Additionally, please clarify whether survey weights and clustering were fully accounted for in all regression analyses.

Response: Thank you for the recommendation. We agree that several modeling strategies may be used for skewed outcomes. However, our research question focused on estimating the association between RFM and the Framingham Risk Score. For this purpose, a generalized linear model with a gamma family and log link was selected because it models the conditional mean of a positively skewed continuous outcome and provides multiplicative, population-averaged estimates, which are appropriate for addressing this type of association. Quantile regression, while valuable for examining how an exposure affects specific quantiles of the outcome distribution, addresses a different scientific question and is not designed to estimate mean associations. Because our study did not aim to investigate distributional differences across percentiles, quantile regression was not the optimal analytic approach. Additionally, for quantile regression STATA statistical package does not account for the complex survey weights. This would yield imprecise estimates given the national survey we used, and thus represents a limitation.

Regarding the use of survey weights and the complex survey design in regression models we would like to note that in the submitted manuscript we declared in the Statistical Analysis section that “Given the characteristics of the VIANEV survey, the analyses accounted for sample weights and the complex survey design”. Furthermore, for further clarity Table 3 has a footnote stating this, and it reads “All estimates accounted for the VIANEV sample design”

Changes: None

6. Potential Confounders

-Important confounders such as alcohol consumption, physical activity beyond sedentary time, and dietary quality were not included. The absence of these variables should be acknowledged as a limitation in more detail.

Response: Thank you for the observation. When selecting potential confounding variables for our models, we considered including alcohol consumption, which is available in the dataset (alcohol use during the past 30 days). Initially, we decided not to include it because our primary focus was on variables most directly related to adiposity and cardiometabolic risk; however, we agree that alcohol consumption is an important behavioral factor. Based on your suggestion, we have now incorporated alcohol consumption into all adjusted models (see revised Tables 1, 2, and 3). After inclusion, the coefficients remained virtually unchanged, with only one estimate showing a minimal increase (from 0.47 [0.32–0.63] to 0.48 [0.32–0.63]), indicating that alcohol consumption did not materially alter the associations.

Regarding physical activity, we opted to adjust for sedentary time because prolonged sitting is independently associated with adverse cardiometabolic outcomes, even among individuals who meet physical activity recommendations (1). Sedentary behavior therefore captures a distinct and relevant dimension of movement-related exposure. Nonetheless, we acknowledge that the absence of more detailed physical activity indicators may result in residual confounding. No dietary quality data was available for which we did not include this as a confounding variable.

1. Liang, Zhi-de et al. “Association between sedentary behavior, physical activity, and cardiovascular disease-related outcomes in adults-A meta-analysis and systematic review.” Frontiers in public health vol. 10 1018460. 19 Oct. 2022, doi:10.3389/fpubh.2022.1018460

Changes: Following your recommendation, we have added the following to our limitations section:

“The lack of potentially confounding variables, such as diet quality, energy intake or more specific physical activity measurement, could also result in residual confounding, affecting the magnitude and direction of the estimated associations due to its strong influence on body fat percentage and components of the Framingham Risk Score; nevertheless, the adjustment models included indirect measures of diet quality (daily consumption of five or more portions of fruits and vegetables) and physical activity (daily hours of sedentary behavior) in order to reduce confounding.”

7. Sex-Specific Findings

- The stronger association in women is intriguing. The Discussion offers some explanations (biological and social), but these remain speculative. The authors should expand on possible mechanisms and highlight the need for future longitudinal research to clarify causality.

Response: Thank you for the observation. We agree with the reviewer’s observation and have incorporated a text addressing this topic.

Changes: The following text was added in the discussion section:

“Additionally, our findings show a stronger association between obesity and cardiovascular risk in women, which is consistent with evidence indicating a higher burden of cardiovascular risk factors in women compared to men [38]. In Peru, studies have similarly reported a higher prevalence of obesity and metabolic syndrome among women [39,40]. Likewise, an analysis of the nationally representative ENDES 2019 survey found that women had markedly higher odds of abdominal obesity than men, reinforcing the notion of sex-specific vulnerability in adiposity-related cardiometabolic risk [40]. Together, these findings provide context for why increases in RFM may correspond to larger relative changes in predicted 10-year cardiovascular risk among women in our study.

In line with this, biological factors, such as greater susceptibility to chronic diseases and the interaction between estrogens and age, as well as social factors including disparities in the detection and treatment of CVD, could also explain women’s higher vulnerability [38]. On the other hand, epidemiological studies in Latin America [4] have reported a higher incidence of CVD in men than in women (655 vs. 531 per 100,000 individuals), and additional evidence indicates that Peruvian men present worse cardiovascular health than women [5,41]. Of note, compared with previous re

---

## [Decision Letter · Decision Letter 1]

4 Jan 2026

Dear Dr. Hernández-Vásquez,

Thank you for submitting your manuscript to PLOS ONE. After careful consideration, we feel that it has merit but does not fully meet PLOS ONE’s publication criteria as it currently stands. Therefore, we invite you to submit a revised version of the manuscript that addresses the points raised during the review process.

We look forward to receiving your revised manuscript.

Kind regards,

Neftali Eduardo Antonio-Villa, MD PhD

Academic Editor

PLOS One

Journal Requirements:

Reviewers' comments:

Reviewer's Responses to Questions

**Comments to the Author**

Reviewer #1: All comments have been addressed

2. Is the manuscript technically sound, and do the data support the conclusions?

Reviewer #1: Yes

3. Has the statistical analysis been performed appropriately and rigorously?

Reviewer #1: I Don't Know

4. Have the authors made all data underlying the findings in their manuscript fully available?

Reviewer #1: (No Response)

5. Is the manuscript presented in an intelligible fashion and written in standard English?

Reviewer #1: Yes

Reviewer #1: 1. Novelty and Framing of Contribution

Although the authors now acknowledge ethnic and body composition differences in the Introduction, the unique contribution of this study remains under-emphasized.

- The manuscript would benefit from a clearer statement of contribution beyond “first in Peru.”

- Specifically, the authors should emphasize:

- The comparison of multiple operationalizations of RFM (binary, continuous, tertiles)

- The sex-stratified modeling with complex survey adjustment

Recommendation:

Add 1–2 sentences in the final paragraph of the Introduction explicitly stating how this study advances knowledge beyond prior international RFM studies.

2. Use of the Framingham Risk Score

The authors appropriately acknowledge the lack of local validation and justify the use of the Framingham score due to Ministry of Health endorsement. However:

- The manuscript still risks being interpreted as estimating true cardiovascular risk rather than relative predicted risk.

- Some phrasing in the Results and Discussion (e.g., “higher cardiovascular risk”) could be misinterpreted by readers unfamiliar with risk score limitations.

Recommendation:

Systematically use wording such as “higher estimated 10-year cardiovascular risk” or “higher Framingham risk score” throughout the Results and Discussion to avoid overinterpretation.

3. Interpretation of Effect Sizes from Gamma GLM

The use of β coefficients from a log-linked gamma model is statistically sound, but interpretability remains limited for non-technical readers.

- While the authors chose not to report prevalence ratios, readers may still struggle to understand the clinical relevance of coefficients such as β = 0.48.

- The newly added absolute mean differences are helpful but underutilized.

Recommendation:

In the Discussion, briefly interpret at least one adjusted estimate in plain language (e.g., relative or proportional increase in Framingham score) to improve accessibility.

4. Sex-Specific Findings

The expanded discussion on sex differences is thoughtful and well-referenced. However:

- The section is now quite long and diffuse, mixing Peruvian data, regional incidence, and methodological distinctions between predicted risk and observed events.

- Some arguments appear defensive rather than explanatory.

Recommendation:

Condense this section slightly and clearly separate:

1.Biological/social explanations

2.Differences between predicted risk and observed incidence

3.Implications for future longitudinal research

This will improve coherence and readability.

**Do you want your identity to be public for this peer review?** For information about this choice, including consent withdrawal, please see our Privacy Policy

Reviewer #1: No

---

## [Author Response · Author response to Decision Letter 2]

6 Jan 2026

Response Letter - Round 2

Reviewer #1:

1. Novelty and Framing of Contribution

Although the authors now acknowledge ethnic and body composition differences in the Introduction, the unique contribution of this study remains under-emphasized.

- The manuscript would benefit from a clearer statement of contribution beyond “first in Peru.”

- Specifically, the authors should emphasize:

- The comparison of multiple operationalizations of RFM (binary, continuous, tertiles)

- The sex-stratified modeling with complex survey adjustment

Recommendation:

Add 1–2 sentences in the final paragraph of the Introduction explicitly stating how this study advances knowledge beyond prior international RFM studies.

Response:

We thank the reviewer for this important observation. We agree that our contribution extends beyond being the first study in Peru and should highlight the methodological advances and analytical approach that distinguish our work from previous international studies. We have revised the final paragraph of the Introduction to better articulate these specific contributions.

Change:

Now the text reads: Therefore, this study aimed to evaluate the association between obesity defined by RFM, considered under different operationalization approaches (binary, continuous, and tertile-based) and 10-year cardiovascular risk stratified by sex, as estimated by the Framingham score in Peruvian adults who participated in a national survey during 2017–2018. The present study advances knowledge beyond prior international RFM studies by comparing multiple operationalizations of RFM (binary, continuous, and tertiles) and applying sex-stratified analyses within a complex survey design in a nationally representative Latin American population, providing novel insights into optimal analytical strategies for RFM-cardiovascular risk assessment in ethnically diverse contexts.

2. Use of the Framingham Risk Score

The authors appropriately acknowledge the lack of local validation and justify the use of the Framingham score due to Ministry of Health endorsement. However:

- The manuscript still risks being interpreted as estimating true cardiovascular risk rather than relative predicted risk.

- Some phrasing in the Results and Discussion (e.g., “higher cardiovascular risk”) could be misinterpreted by readers unfamiliar with risk score limitations.

Recommendation:

Systematically use wording such as “higher estimated 10-year cardiovascular risk” or “higher Framingham risk score” throughout the Results and Discussion to avoid overinterpretation.

Response:

We thank the reviewer for this observation. We fully agree that our terminology must clearly distinguish between predicted risk scores and actual cardiovascular risk, especially given the lack of local validation of the Framingham score in Peruvian populations.

Change:

We have made changes to the Results and Discussion sections to explicitly specify that the outcome refers to “risk.”

3. Interpretation of Effect Sizes from Gamma GLM

The use of β coefficients from a log-linked gamma model is statistically sound, but interpretability remains limited for non-technical readers.

- While the authors chose not to report prevalence ratios, readers may still struggle to understand the clinical relevance of coefficients such as β = 0.48.

- The newly added absolute mean differences are helpful but underutilized.

Recommendation:

In the Discussion, briefly interpret at least one adjusted estimate in plain language (e.g., relative or proportional increase in Framingham score) to improve accessibility.

Response:

We thank the reviewer for this observation. We agree with the suggestion and have added a plain-language interpretation in the first paragraph of the Discussion section.

Change:

The objective of the present study was to evaluate the association between obesity defined by RFM and the estimated 10-year cardiovascular risk according to estimated through the Framingham risk score in Peruvian adults aged 30 to 59 years. A positive and significant association was observed in both sexes. Furthermore, it was found that for each unit increase in RFM, the estimated cardiovascular risk Framingham risk score increased by 0.07 in men and 0.06 in women, corresponding to approximately 7% and 6% higher estimated risk, respectively. This indicates that higher estimated total body fat percentage is associated with proportionally higher estimated 10-year cardiovascular risk indicating that for each increase in the estimated total body fat percentage, the estimated 10-year cardiovascular risk also increases. To the best of our knowledge, this is the first study to evaluate the association between RFM, an alternative estimator of total body fat percentage, and the estimated 10-year cardiovascular risk using the Framingham risk score.

4. Sex-Specific Findings

The expanded discussion on sex differences is thoughtful and well-referenced. However:

- The section is now quite long and diffuse, mixing Peruvian data, regional incidence, and methodological distinctions between predicted risk and observed events.

- Some arguments appear defensive rather than explanatory.

Recommendation:

Condense this section slightly and clearly separate:

1.Biological/social explanations

2.Differences between predicted risk and observed incidence

3.Implications for future longitudinal research

This will improve coherence and readability.

Response:

We thank the reviewer for this observation. We have thoroughly reviewed the content of the Discussion section, which was was refined and reorganized to improve coherence and readability with added subheadings to facilitate identification of the following sections:

1) Comparison with previous studies

2) Potential explanatory mechanisms

3) Sex-based differences

4) Implications for public health and clinical practice

5) Study limitations.

Changes: The discussion section has been revised. The sex-specific findings section was refined and reorganized to improve clarity.

---

## [Decision Letter · Decision Letter 2]

22 Jan 2026

Dear Dr. Hernández-Vásquez,

Thank you for submitting your manuscript to PLOS ONE. After careful consideration, we feel that it has merit but does not fully meet PLOS ONE’s publication criteria as it currently stands. Therefore, we invite you to submit a revised version of the manuscript that addresses the points raised during the review process.

We look forward to receiving your revised manuscript.

Kind regards,

Neftali Eduardo Antonio-Villa, MD PhD

Academic Editor

PLOS One

Journal Requirements:

Additional Editor Comments:

Please address the comments raised by the reviewer. In particular, emphasize the consistency of the findings between the abstract and conclusions, justify the selection of confounders, and soften the tone of any inferential conclusions.

Reviewers' comments:

Reviewer's Responses to Questions

**Comments to the Author**

Reviewer #1: All comments have been addressed

2. Is the manuscript technically sound, and do the data support the conclusions?

Reviewer #1: Yes

3. Has the statistical analysis been performed appropriately and rigorously?

Reviewer #1: Yes

4. Have the authors made all data underlying the findings in their manuscript fully available?

Reviewer #1: Yes

5. Is the manuscript presented in an intelligible fashion and written in standard English?

Reviewer #1: Yes

Reviewer #1: Comments

1. Clarification of Outcome Interpretation

The authors have appropriately revised much of the text to emphasize “estimated 10-year cardiovascular risk” rather than true event risk. However, a small number of phrases still risk causal or clinical overinterpretation, particularly in the Abstract and Conclusion where statements such as “higher cardiovascular risk” appear without explicit reference to prediction.

Recommendation:

- Perform a final consistency check across the Abstract, Results, Discussion, and Conclusion to ensure uniform use of terms such as “higher Framingham risk score” or “higher estimated 10-year cardiovascular risk.”

This will further reduce the risk of misinterpretation by non-specialist readers.

2. Interpretation of Effect Sizes in Log-Gamma Models

The added plain-language interpretation of the continuous RFM coefficients (e.g., ~6–7% higher estimated risk per unit increase) is a valuable improvement. However, the interpretation currently appears only once and is not clearly tied to the categorical (binary and tertile-based) analyses.

Recommendation:

- Briefly contextualize at least one categorical comparison (e.g., obese vs. non-obese or highest vs. lowest tertile) in relative or proportional terms in the Discussion.

This would enhance accessibility for readers unfamiliar with generalized linear models while preserving statistical rigor.

3. Cross-sectional Design and Temporality

Although the limitations section appropriately acknowledges the cross-sectional nature of the data, some statements in the Discussion (particularly in the mechanistic and public health implications sections) still imply directionality between adiposity and cardiovascular risk.

Recommendation:

- Slightly temper causal language when referring to mechanisms or implications (e.g., “may contribute to” rather than “leads to”).

- Explicitly reiterate that associations reflect concurrent relationships with predicted risk rather than disease progression.

4. Sex-Stratified Analyses

The reorganization of the Discussion into sub-sections has improved clarity. The section on sex-based differences is now more balanced and less defensive. However, it remains relatively long compared to other sections.

Recommendation:

- Consider further condensation by reducing repetition between regional epidemiology and methodological explanations of predicted versus observed risk.

5. Covariate Selection and Residual Confounding

The rationale for covariate selection is reasonable and consistent with epidemiological standards. However, diet and physical activity are only indirectly captured.

Recommendation:

- Add a brief sentence in the Methods or Limitations explicitly stating that residual confounding related to unmeasured dietary quality and physical activity intensity cannot be excluded.

6. Presentation of Tables

Tables are generally clear and informative. However, Table 2 is dense and may be difficult to interpret for readers.

Recommendation:

- Consider adding a brief interpretive sentence in the Results highlighting the most salient gradients (e.g., age and altitude trends) rather than relying solely on tabular detail.

**Do you want your identity to be public for this peer review?** For information about this choice, including consent withdrawal, please see our Privacy Policy

Reviewer #1: No

---

## [Author Response · Author response to Decision Letter 3]

28 Jan 2026

Response Letter - Round 3

Reviewer #1:

1. Clarification of Outcome Interpretation

The authors have appropriately revised much of the text to emphasize “estimated 10-year cardiovascular risk” rather than true event risk. However, a small number of phrases still risk causal or clinical overinterpretation, particularly in the Abstract and Conclusion where statements such as “higher cardiovascular risk” appear without explicit reference to prediction.

Recommendation:

- Perform a final consistency check across the Abstract, Results, Discussion, and Conclusion to ensure uniform use of terms such as “higher Framingham risk score” or “higher estimated 10-year cardiovascular risk.”

This will further reduce the risk of misinterpretation by non-specialist readers.

Response: We thank the reviewer for this observation.

Changes: Revisions have been made throughout the manuscript in accordance with the recommendations.

2. Interpretation of Effect Sizes in Log-Gamma Models

The added plain-language interpretation of the continuous RFM coefficients (e.g., ~6–7% higher estimated risk per unit increase) is a valuable improvement. However, the interpretation currently appears only once and is not clearly tied to the categorical (binary and tertile-based) analyses.

Recommendation:

- Briefly contextualize at least one categorical comparison (e.g., obese vs. non-obese or highest vs. lowest tertile) in relative or proportional terms in the Discussion.

This would enhance accessibility for readers unfamiliar with generalized linear models while preserving statistical rigor.

Response: Thank you for the clarification. We have incorporated these recommendations into the Discussion section.

Changes: We added interpretations in proportional terms, as well as interpretations of the results stratified by tertiles.

3. Cross-sectional Design and Temporality

Although the limitations section appropriately acknowledges the cross-sectional nature of the data, some statements in the Discussion (particularly in the mechanistic and public health implications sections) still imply directionality between adiposity and cardiovascular risk.

Recommendation:

- Slightly temper causal language when referring to mechanisms or implications (e.g., “may contribute to” rather than “leads to”).

- Explicitly reiterate that associations reflect concurrent relationships with predicted risk rather than disease progression.

Response: We thank the reviewer for this observation. We revised the Discussion to further temper causal language, particularly in the sections addressing potential mechanisms and public health implications.

Changes: Revisions have been made throughout the manuscript in accordance with the recommendations. Relevant statements were rephrased to emphasize that our findings reflect concurrent associations between RFM-defined adiposity and estimated 10-year cardiovascular risk, rather than causal effects or disease progression.

4. Sex-Stratified Analyses

The reorganization of the Discussion into sub-sections has improved clarity. The section on sex-based differences is now more balanced and less defensive. However, it remains relatively long compared to other sections.

Recommendation:

- Consider further condensation by reducing repetition between regional epidemiology and methodological explanations of predicted versus observed risk.

Response: Thank you for the clarification. We agree that this section is relatively lengthy.

Change: We have synthesized the main ideas of this subsection while preserving the key message we aim to convey to readers.

5. Covariate Selection and Residual Confounding

The rationale for covariate selection is reasonable and consistent with epidemiological standards. However, diet and physical activity are only indirectly captured.

Recommendation:

- Add a brief sentence in the Methods or Limitations explicitly stating that residual confounding related to unmeasured dietary quality and physical activity intensity cannot be excluded.

Response: Thank you for the clarification. We agree that it is important to explicitly state that adjustment for these variables does not exclude residual confounding.

Changes: We have clarified this in the Limitations section.

6. Presentation of Tables

Tables are generally clear and informative. However, Table 2 is dense and may be difficult to interpret for readers.

Recommendation:

- Consider adding a brief interpretive sentence in the Results highlighting the most salient gradients (e.g., age and altitude trends) rather than relying solely on tabular detail.

Response: Thank you for the clarification regarding synthesizing the findings of Table 2 in the Results section.

Changes: We have made changes to the interpretation of the findings.

---

## [Editor Report · Decision Letter 3]

1 Feb 2026

Relative fat mass and cardiovascular risk in Peruvian adults: Findings from a national survey

PONE-D-25-46369R3

Dear Dr. Hernández-Vásquez,

We’re pleased to inform you that your manuscript has been judged scientifically suitable for publication and will be formally accepted for publication once it meets all outstanding technical requirements.

Kind regards,

Neftali Eduardo Antonio-Villa, MD PhD

Academic Editor

PLOS One

Additional Editor Comments (optional):

Thank you for addressing all of the referees’ comments. I read the revised manuscript with great interest, and I believe it fills an important research gap in Peru. I congratulate the authors and look forward to seeing future work on this topic from the study group.
---

## [Editor Report · Acceptance letter]

PONE-D-25-46369R3

PLOS One

Dear Dr. Hernández-Vásquez,

I'm pleased to inform you that your manuscript has been deemed suitable for publication in PLOS One. Congratulations! Your manuscript is now being handed over to our production team.

Kind regards,

on behalf of

Dr. Neftali Eduardo Antonio-Villa

Academic Editor

PLOS One